# CORRELATED VARIATIONAL AUTO-ENCODERS

**Da Tang** *
Columbia University
datang@cs.columbia.edu

**Dawen Liang**
Netflix Inc.
dliang@netflix.com

**Tony Jebara**
Columbia University & Netflix Inc.
jebara@cs.columbia.edu

**Nicholas Ruozzi**
University of Texas at Dallas
nicholas.ruozzi@utdallas.edu

## ABSTRACT

Variational Audo-Encoders (VAEs) are capable of learning latent representations for high dimensional data. However, due to the i.i.d. assumption, VAEs only optimize the singleton variational distributions and fail to account for the correlations between data points, which might be crucial for learning latent representations from dataset where *a priori* we know correlations exist. We propose Correlated Variational Auto-Encoders (CVAEs) that can take the correlation structure into consideration when learning latent representations with VAEs. CVAEs apply a prior based on the correlation structure. To address the intractability introduced by the correlated prior, we develop an approximation by the average of a set of tractable lower bounds over all *maximal acyclic subgraphs* of the undirected correlation graph. Experimental results on matching and link prediction on public benchmark rating datasets and spectral clustering on a synthetic dataset show the effectiveness of the proposed method over baseline algorithms.

## 1 INTRODUCTION

Variational Auto-Encoders (VAEs) (Kingma & Welling, 2014; Rezende et al., 2014) are a family of powerful deep generative models that learns stochastic latent embeddings for input data. By applying variational inference on deep generative models, VAEs are able to successfully identify the latent structures of the data and learn latent distributions that can potentially extract more compact information that is not easily and directly obtained from the original data.

VAEs assume each data point is i.i.d. generated, which means we do not consider any correlations between the data points. This is a reasonable assumption under many settings. However, sometimes we know *a priori* that data points are correlated, e.g., in graph-structured datasets (Bruna et al., 2015; Shi et al., 2014; Kipf & Welling, 2016; Hamilton et al., 2017) . In these cases, it is more reasonable to assume the latent representation for each data point also respects the correlation graph structure.

In this paper, we extend the standard VAEs by encouraging the latent representations to take the correlation structure into account, which we term CVAEs. In CVAEs, rather than a commonly used i.i.d. standard Gaussian prior, we apply a correlated prior on the latent variables following the structure known *a priori*. We develop two variations, CVAE$_{ind}$ and CVAE$_{corr}$: In CVAE$_{ind}$, we still use a fully-factorized singleton variational density via amortized inference; while in CVAE$_{corr}$, instead of only learning singleton latent variational density functions, we also incorporate pairwise latent variational density functions to achieve a better variational approximation. With a correlated prior, the standard variational inference objective becomes intractable to compute. We sidestep the challenging objective by considering the average of a set of tractable lower bounds over all *maximal acyclic subgraphs* of the given undirected correlation graph.

The experimental results show that both CVAE$_{ind}$ and CVAE$_{corr}$ can outperform the baseline methods on three tasks: matching dual user pairs using the rating records on a movie recommendation dataset,

---

*Most of this work was done when Da Tang was an intern at Netflix.

clustering the vertices with latent embeddings drawn from a tree-structured undirected graphical model on a synthetic dataset, and predicting links using the rating records on a product rating dataset.

## 2 CVAES ON ACYCLIC GRAPHS

In this section, we extends VAEs to fit with a set of latent variables with an acyclic correlation graph. We start with acyclic graphs since the prior and posterior approximation of the latent variables can be expressed exactly for such correlation structures.

### 2.1 VARIATIONAL AUTO-ENCODINGS

Assume that we have input data $\boldsymbol{x} = \{\boldsymbol{x}_1, \ldots, \boldsymbol{x}_n\} \subseteq \mathbb{R}^D$. Standard VAEs assume that the data point $\boldsymbol{x}_i$ is generated i.i.d. from the following process: First, generate the latent variables $\boldsymbol{z} = \{\boldsymbol{z}_1, \ldots, \boldsymbol{z}_n\} \subseteq \mathbb{R}^d$ (usually $d \ll D$) by drawing $\boldsymbol{z}_i \overset{i.i.d.}{\sim} p_0(\boldsymbol{z}_i)$ from the prior distribution $p_0$ (parameter-free, usually a standard Gaussian distribution) for each $i \in \{1, \ldots, n\}$. Then generate the data points $\boldsymbol{x}_i \sim p_{\boldsymbol{\theta}}(\boldsymbol{x}_i|\boldsymbol{z}_i)$ from the model conditional distribution $p_{\boldsymbol{\theta}}$, for $i \in \{1, \ldots, n\}$ independently.

We are interested in optimizing $\boldsymbol{\theta}$ to maximize the likelihood $p_{\boldsymbol{\theta}}(\boldsymbol{x})$, which requires computing the posterior distribution $p_{\boldsymbol{\theta}}(\boldsymbol{z}|\boldsymbol{x}) = \prod_{i=1}^{n} p_{\boldsymbol{\theta}}(\boldsymbol{z}_i|\boldsymbol{x}_i)$. For most models, this is usually intractable. VAEs sidestep the intractability and resort to variational inference by approximating this posterior distribution as $q_{\boldsymbol{\lambda}}(\boldsymbol{z}|\boldsymbol{x}) = \prod_{i=1}^{n} q_{\boldsymbol{\lambda}}(\boldsymbol{z}_i|\boldsymbol{x}_i)$ via amortized inference and maximize the *evidence lower bound* (ELBO):

$$\begin{aligned}
\mathcal{L}(\boldsymbol{\lambda}, \boldsymbol{\theta}) &= \mathbb{E}_{q_{\boldsymbol{\lambda}}(\boldsymbol{z}|\boldsymbol{x})}\left[\log p_{\boldsymbol{\theta}}(\boldsymbol{x}|\boldsymbol{z})\right] - \mathrm{KL}(q_{\boldsymbol{\lambda}}(\boldsymbol{z}|\boldsymbol{x})||p_0(\boldsymbol{z})) \\
&= \sum_{i=1}^{n}(\mathbb{E}_{q_{\boldsymbol{\lambda}}(\boldsymbol{z}_i|\boldsymbol{x}_i)}\left[\log p_{\boldsymbol{\theta}}(\boldsymbol{x}_i|\boldsymbol{z}_i)\right] - \mathrm{KL}(q_{\boldsymbol{\lambda}}(\boldsymbol{z}_i|\boldsymbol{x}_i)||p_0(\boldsymbol{z}_i))).
\end{aligned} \tag{1}$$

The ELBO is a lower bounds to the log-likelihood $\log p_{\boldsymbol{\theta}}(\boldsymbol{x})$, and maximizing this lower bound is equivalent to minimizing the KL-divergence between the variational distribution $q_{\boldsymbol{\lambda}}(\boldsymbol{z}|\boldsymbol{x})$ and the true posterior $p_{\boldsymbol{\theta}}(\boldsymbol{z}|\boldsymbol{x})$. The KL-divergence term in the ELBO can be viewed as regularization that pulls the variational distribution $q_{\boldsymbol{\lambda}}(\boldsymbol{z}|\boldsymbol{x})$ towards the prior $p_0(\boldsymbol{z})$. Since the approximation family $q_{\boldsymbol{\lambda}}(\boldsymbol{z}|\boldsymbol{x})$ factorizes over data points and the prior is i.i.d. Gaussian, the KL-divergence in the ELBO is simply a sum over the per-data-point KL-divergence terms, which means that we do not consider any correlations of latent representation between data points.

### 2.2 CORRELATED PRIORS ON ACYCLIC GRAPHS

As motivated earlier, sometimes we know *a priori* that there exist correlations between data points. If we have access to such information, we can incorporate it into the generative process of VAEs, giving us correlated VAEs.

Formally, assume that we have $n$ data points $\boldsymbol{x}_1, \ldots, \boldsymbol{x}_n$. In addition, we assume that the correlation structure of these data points is given by an undirected graph $G = (V, E)$, where $V = \{v_1, ..., v_n\}$ is the set of vertices corresponding to all data points (i.e., $v_i$ corresponds to $\boldsymbol{x}_i$) and $(v_i, v_j) \in E$ if $\boldsymbol{x}_i$ and $\boldsymbol{x}_j$ are correlated. For now we assume $G$ is acyclic, and later in Section 3 we will extend the results to general graphs. Making use of the correlation information, we change the prior distribution $p_0^{\mathrm{corr}}$ of the latent variables $\boldsymbol{z}_1, \ldots, \boldsymbol{z}_n$ to take the form of a distribution over $(\boldsymbol{z}_1, \ldots, \boldsymbol{z}_n) \in \mathbb{R}^d \times \ldots \times \mathbb{R}^d$ whose singleton and pairwise marginal distributions satisfy

$$p_0^{\mathrm{corr}}(\boldsymbol{z}_i) = p_0(\boldsymbol{z}_i) \text{ for all } v_i \in V, \quad p_0^{\mathrm{corr}}(\boldsymbol{z}_i, \boldsymbol{z}_j) = p_0(\boldsymbol{z}_i, \boldsymbol{z}_j) \text{ if } (v_i, v_j) \in E. \tag{2}$$

Here $p_0(\cdot)$ is a parameter-free density function that captures the singleton prior distribution and $p_0(\cdot, \cdot)$ is a parameter-free density function that captures the pairwise correlation between each pair of variables. Furthermore, they should satisfy the following symmetry and marginalization consistency properties:

$$p_0(\boldsymbol{z}_i, \boldsymbol{z}_j) = p_0(\boldsymbol{z}_j, \boldsymbol{z}_i) \text{ for all } \boldsymbol{z}_i, \boldsymbol{z}_j \in \mathbb{R}^d, \quad \int p_0(\boldsymbol{z}_i, \boldsymbol{z}_j)d\boldsymbol{z}_j = p_0(\boldsymbol{z}_i) \text{ for all } \boldsymbol{z}_i \in \mathbb{R}^d. \tag{3}$$

The symmetry property (the first part of Eq. 3) preserves the validity of the pairwise marginal distributions since $p_0(\boldsymbol{z}_i, \boldsymbol{z}_j)$ and $p_0(\boldsymbol{z}_j, \boldsymbol{z}_i)$ are representing exactly the same distribution. The marginalization consistency property (the second part of Eq. 3) maintains the consistency between the singleton and pairwise density functions.

This prior will help the model take the correlation information into consideration since we have a KL-divergence regularization term in the ELBO that will push the variational distribution towards the prior distribution. With the correlated prior defined above, the generative process of a CVAE is straightforward: First we sample $\boldsymbol{z}$ from this new prior $p_0^{\text{corr}}$, then we sample each data point $\boldsymbol{x}_i$ conditionally independently from $\boldsymbol{z}_i$, similar to a standard VAE.

In general, the prior distributions defined in Eqs. 2 and 3 do not necessarily form a valid joint distribution on $\boldsymbol{z}$, that is, there exists a graph $G$ and choice of prior distributions as above such that no joint distribution on $\boldsymbol{z}$ has the priors defined by $p_0$ as its marginals. Nonetheless, if $G$ is an undirected acyclic graph, such a joint distribution does exist, independent of which distributions we choose for $p_0(\cdot)$ and $p_0(\cdot, \cdot)$ (Wainwright & Jordan, 2008):

$$p_0^{\text{corr}}(\boldsymbol{z}) = \prod_{i=1}^{n} p_0(\boldsymbol{z}_i) \prod_{(v_i, v_j) \in E} \frac{p_0(\boldsymbol{z}_i, \boldsymbol{z}_j)}{p_0(\boldsymbol{z}_i) p_0(\boldsymbol{z}_j)}. \tag{4}$$

In other words, the joint correlated prior distribution on $\boldsymbol{z}$ can be expressed explicitly with only the singleton and pairwise marginal distributions, without having an intractable normalization constant.

## 2.3 CVAEs on acyclic graphs

To perform variational inference, we need to specify which variational family we will use. In this paper, we explore two different variational families: one where the prior distribution only involves singleton marginals, which we denote CVAE$_{\text{ind}}$, and one in which the prior distribution has the form in Eq. 4, which we denote CVAE$_{\text{corr}}$.

**Singleton variational family.** In CVAE$_{\text{ind}}$, we approximate the posterior distribution $p(\boldsymbol{z}|\boldsymbol{x})$ as $q_{\boldsymbol{\lambda}}(\boldsymbol{z}|\boldsymbol{x}) = \prod_{i=1}^{n} q_{\boldsymbol{\lambda}}(\boldsymbol{z}_i|\boldsymbol{x}_i)$ which consists of fully-factorized distributions. The corresponding ELBO for CVAE$_{\text{ind}}$ on acyclic graphs can be found in Appendix A. Even though this variational family does not consider pairwise correlations, the prior $p_0^{\text{corr}}$ is still more expressive than the standard Gaussian prior used in standard VAEs.

**Correlated variational family.** In CVAE$_{\text{corr}}$, where the prior $p_0^{\text{corr}}$ can be written as in Eq. 4, it makes sense to use a variational distribution $q_{\boldsymbol{\lambda}}(\boldsymbol{z} \mid \boldsymbol{x})$ expressed in the same form, i.e., with only singleton and pairwise marginal distributions:

$$q_{\boldsymbol{\lambda}}(\boldsymbol{z} \mid \boldsymbol{x}) = \prod_{i=1}^{n} q_{\boldsymbol{\lambda}}(\boldsymbol{z}_i|\boldsymbol{x}_i) \prod_{(v_i, v_j) \in E} \frac{q_{\boldsymbol{\lambda}}(\boldsymbol{z}_i, \boldsymbol{z}_j|\boldsymbol{x}_i, \boldsymbol{x}_j)}{q_{\boldsymbol{\lambda}}(\boldsymbol{z}_i|\boldsymbol{x}_i) q_{\boldsymbol{\lambda}}(\boldsymbol{z}_j|\boldsymbol{x}_j)}. \tag{5}$$

As in Eq. 3, the marginals need to satisfy the following properties:

$$\begin{cases} q_{\boldsymbol{\lambda}}(\boldsymbol{z}_i, \boldsymbol{z}_j|\boldsymbol{x}_i, \boldsymbol{x}_j) = q_{\boldsymbol{\lambda}}(\boldsymbol{z}_j, \boldsymbol{z}_i|\boldsymbol{x}_j, \boldsymbol{x}_i) & \text{for all } \boldsymbol{z}_i, \boldsymbol{z}_j, \boldsymbol{x}_i, \boldsymbol{x}_j, \\ \int q_{\boldsymbol{\lambda}}(\boldsymbol{z}_i, \boldsymbol{z}_j|\boldsymbol{x}_i, \boldsymbol{x}_j) d\boldsymbol{z}_j = q_{\boldsymbol{\lambda}}(\boldsymbol{z}_i|\boldsymbol{x}_i) & \text{for all } \boldsymbol{z}_i, \boldsymbol{x}_i, \boldsymbol{x}_j. \end{cases}$$

The corresponding ELBO for CVAE$_{\text{corr}}$ on acyclic graphs can be found in Appendix A. This ELBO yields a tighter lower bound on the log-likelihood $\log p_\theta(\boldsymbol{x})$ under the prior in Eq. 4 as compared to the ELBO of CVAE$_{\text{ind}}$, since optimizing this ELBO involves with optimizing over a larger set of distributions than the factorized distributions as in CVAE$_{\text{ind}}$. CVAE$_{\text{corr}}$ not only takes the correlation structure into consideration as CVAE$_{\text{ind}}$, but also learns correlated variational distributions that can potentially yield better approximations to the model posterior.

## 3 CVAEs on general graphs

We have introduced the CVAEs for acyclic correlation structures. In this section, we extend these models to general undirected graphs.

### 3.1 WHY THE TRIVIAL GENERALIZATION FAILS

A simple generalization is just to directly apply the ELBOs (shown in Appendix A) for acyclic graphs that we developed in Section 2. However, this simple approach can easily fail. First, as described earlier, Eq. 4 is not guaranteed to be a valid distribution (i.e., it does not integrate to 1 over $z$) for a general graph $G$ (Wainwright & Jordan, 2008). Moreover, optimizing these acyclic ELBOs may lead to pathological cases where the objectives go to infinity due to these objectives not guaranteed to be a lower bound of the log-likelihood for general graphs. We construct a simple example to demonstrate this in Appendix B. Therefore, we can not directly apply the ELBOs on acyclic graphs. We need to define a new prior distribution $p_0^{\mathrm{corr}_g}(z)$ and derive a new lower bound for the log-likelihood $\log p_\theta(x)$ under this prior, for general graphs.

### 3.2 INFERENCE WITH A WEIGHTED OBJECTIVE

Though a general graph $G$ may contain cycles, its acyclic subgraphs may contain correlation structures that well-approximate it, especially when $G$ is sparse. For acyclic graphs, we have already derived ELBOs for CVAE$_{\mathrm{ind}}$ and CVAE$_{\mathrm{corr}}$ in Section 2. We extend these two methods for general graph $G$ by considering an average of the loss over its "maximal acyclic subgraphs", which is defined as follows:

**Definition 1** (Maximal acyclic subgraph). *For an undirected graph $G = (V, E)$, say an acyclic subgraph $G' = (V', E')$ is a maximal acyclic subgraph of $G$ if:*

- *$V' = V$, i.e., $G'$ contains all vertices of $G$.*

- *Adding any edge from $E/E'$ to $E'$ will create a cycle in $G'$.*

A maximal acyclic subgraph of $G$ may be in similar structure to $G$, especially when $G$ is sparse. When $G$ is acyclic, it only contains one maximal acyclic subgraph (i.e., $G$ itself). When $G$ is connected, any subgraph $G'$ is a maximal acyclic subgraph of $G$ if and only if $G'$ is a spanning tree of $G$. In general, any subgraph $G'$ is a maximal acyclic subgraph of $G$, if and only if, for any of $G$'s connected components, $G'$ contains a spanning tree over this connected component, see Figure 1. We will use $\mathcal{A}_G$ to denote the set of all maximal acyclic subgraphs of $G$.

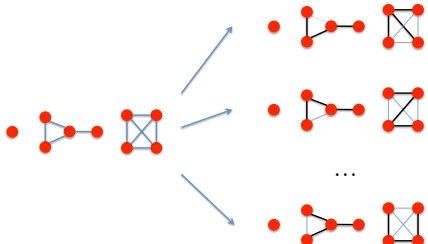

Figure 1: Visualization of the set of maximal acyclic subgraphs (right) of the given graph $G$ (left). On the right, the dark solid edges are selected and light dashed edges not selected. As can be seen from this figure, each subgraph $G' \in \mathcal{A}_G$ is just a combination of a spanning trees over all of $G$'s connected components. In total, this graph $G$ has $|\mathcal{A}_G| = 48$ maximal acyclic subgraphs.

While Eq. 4 is not guaranteed to be a valid prior distribution on $z$ if $G$ contains cycles, we can use it to define a new prior over $G$'s set of maximal acyclic subgraphs. More specifically, we define the prior distribution of $z$ as a uniform mixture over all subgraphs in $\mathcal{A}_G$:

$$p_0^{\mathrm{corr}_g}(z) = \frac{1}{|\mathcal{A}_G|} \sum_{G'=(V,E')\in\mathcal{A}_G} p_0^{G'}(z) \tag{6}$$

where $p_0^{G'}(z) = \prod_{i=1}^{n} p_0(z_i) \prod_{(v_i,v_j)\in E'} \frac{p_0(z_i,z_j)}{p_0(z_i)p_0(z_j)}$. Eq. 6 defines a uniform mixture model over the set of prior distributions $p_0^{G'}$ on the maximal acyclic subgraphs of $G$, so it is also a valid density. Note that this reduces to Section 2 when $G$ is acyclic as $|\mathcal{A}_G| = 1$. Under this definition, the log-likelihood

$\log p_{\boldsymbol{\theta}}(\boldsymbol{x})$ becomes

$$
\begin{aligned}
\log p_{\boldsymbol{\theta}}(\boldsymbol{x}) = \log \mathbb{E}_{p_0^{\mathrm{corr}_g}(\boldsymbol{z})}[p_{\boldsymbol{\theta}}(\boldsymbol{x}|\boldsymbol{z})] &\geq \frac{1}{|\mathcal{A}_G|} \sum_{G' \in \mathcal{A}_G} \mathbb{E}_{p_0^{G'}(\boldsymbol{z})}[\log p_{\boldsymbol{\theta}}(\boldsymbol{x}|\boldsymbol{z})] \\
&\geq \frac{1}{|\mathcal{A}_G|} \sum_{G' \in \mathcal{A}_G} \left( \mathbb{E}_{q_{\boldsymbol{\lambda}}^{G'}(\boldsymbol{z}|\boldsymbol{x})}[\log p_{\boldsymbol{\theta}}(\boldsymbol{x}|\boldsymbol{z})] - \mathrm{KL}(q_{\boldsymbol{\lambda}}^{G'}(\boldsymbol{z}|\boldsymbol{x})||p_0^{G'}(\boldsymbol{z})) \right).
\end{aligned}
\tag{7}
$$

The inequality is a direct application of Jensen's inequality, which is also applied when deriving the normal ELBO in Eq. 1. Here $q_{\boldsymbol{\lambda}}^{G'}(\boldsymbol{z}|\boldsymbol{x})$ is a variational distribution related to the maximal acyclic subgraph $G' = (V, E')$. Similar to Section 2, we can specify either a singleton or a correlated variational family. We describe the construction for correlated variational families as singleton variational families are simply a special case of this. We define $q^{G'}$ the same way as in Eq. 5:

$$
q_{\boldsymbol{\lambda}}^{G'}(\boldsymbol{z}) = \prod_{i=1}^{n} q_{\boldsymbol{\lambda}}(\boldsymbol{z}_i|\boldsymbol{x}_i) \prod_{(v_i, v_j) \in E'} \frac{q_{\boldsymbol{\lambda}}(\boldsymbol{z}_i, \boldsymbol{z}_j|\boldsymbol{x}_i, \boldsymbol{x}_j)}{q_{\boldsymbol{\lambda}}(\boldsymbol{z}_i|\boldsymbol{x}_i) q_{\boldsymbol{\lambda}}(\boldsymbol{z}_j|\boldsymbol{x}_j)}.
\tag{8}
$$

Under this definition, we construct $|\mathcal{A}_G|$ different variational distributions $q_{\boldsymbol{\lambda}}^{G'}$. These distributions may be different from each other due to the difference in graph structure, but they share the same singleton and pairwise marginal density functions $q_{\boldsymbol{\lambda}}(\cdot|\cdot)$ and $q_{\boldsymbol{\lambda}}(\cdot, \cdot|\cdot, \cdot)$ as well as the same set of variational parameters $\boldsymbol{\lambda}$. Though these singleton and pairwise marginal density functions are not guaranteed to form a real joint density of $\boldsymbol{z}$ for the whole graph $G$, we can still guarantee that each of the $q_{\boldsymbol{\lambda}}^{G'}$ is a valid density on $\boldsymbol{z}$. Moreover, these locally-consistent singleton and pairwise density functions will approximate the singleton and pairwise marginal posterior distributions.

With this definition of $q_{\boldsymbol{\lambda}}^{G'}$, the lower bound in Eq. 7 becomes the sum of a set of singleton terms over vertices in $V$ and a set of pairwise terms over edges in $E$. The singleton terms have the same weights on all vertices, while the pairwise terms may have different weights: for each edge $e \in E$, the weight is the fraction of times $e$ appears among all subgraphs in $\mathcal{A}_G$. We define this weight as follows:

**Definition 2** (Maximum acyclic subgraph edge weight). *For an undirected graph $G = (V, E)$ and an edge $e \in E$, define $w_{G,e}^{MAS}$ to be the fraction of $G$'s maximal acyclic subgraphs that contain $e$, i.e., $w_{G,e}^{MAS} := \frac{|\{G' \in \mathcal{A}_G : e \in G'\}|}{|\mathcal{A}_G|}$.*

Since each maximal acyclic subgraph of $G$ is a disjoint union of spanning trees, one per connected component of $G$, we have:

**Proposition 1** (Maximum acyclic subgraph edge weight sum). *For an undirected graph $G = (V, E)$, denote $CC(G)$ as the set of connected components of $G$, we have $\sum_{e \in E} w_{G,e}^{MAS} = |V| - |CC(G)|$.*

Using the edge weights, we can show (details in Appendix A) that the lower bound in Eq. 7 equals

$$
\begin{aligned}
&\sum_{i=1}^{n} \left( \mathbb{E}_{q_{\boldsymbol{\lambda}}(\boldsymbol{z}_i|\boldsymbol{x}_i)} \left[ \log p_{\boldsymbol{\theta}}(\boldsymbol{x}_i|\boldsymbol{z}_i) \right] - \mathrm{KL}(q_{\boldsymbol{\lambda}}(\boldsymbol{z}_i|\boldsymbol{x}_i)||p_0(\boldsymbol{z}_i)) \right) - \sum_{(v_i, v_j) \in E} w_{G,(v_i, v_j)}^{\mathrm{MAS}} \cdot \\
&\left( \mathrm{KL}(q_{\boldsymbol{\lambda}}(\boldsymbol{z}_i, \boldsymbol{z}_j|\boldsymbol{x}_i, \boldsymbol{x}_j)||p_0(\boldsymbol{z}_i, \boldsymbol{z}_j)) - \mathrm{KL}(q_{\boldsymbol{\lambda}}(\boldsymbol{z}_i|\boldsymbol{x}_i)||p_0(\boldsymbol{z}_i)) - \mathrm{KL}(q_{\boldsymbol{\lambda}}(\boldsymbol{z}_j|\boldsymbol{x}_j)||p_0(\boldsymbol{z}_j)) \right) \\
&:= \mathcal{L}^{\mathrm{CVAE_{corr}}}(\boldsymbol{\lambda}, \boldsymbol{\theta}).
\end{aligned}
\tag{9}
$$

Eq. 9 defines a valid lower bound of the log-likelihood $\log p_{\boldsymbol{\theta}}(\boldsymbol{x})$ under the mixture model prior in Eq. 6. We define this as the loss function for CVAE$_{\mathrm{corr}}$ on general graphs. As long as the weights $w^{\mathrm{MAS}}$ are tractable, optimizing this lower bound is tractable. We will show how to compute these weights efficiently in Section 3.3. For CVAE$_{\mathrm{ind}}$, the ELBO $\mathcal{L}^{\mathrm{CVAE_{ind}}}$ is just Eq. 9 except that we change $q_{\boldsymbol{\lambda}}(\boldsymbol{z}_i, \boldsymbol{z}_j|\boldsymbol{x}_i, \boldsymbol{x}_j)$ to be the product of the two singleton density functions $q_{\boldsymbol{\lambda}}(\boldsymbol{z}_i|\boldsymbol{x}_i)$ and $q_{\boldsymbol{\lambda}}(\boldsymbol{z}_j|\boldsymbol{x}_j)$.

### 3.3 COMPUTING THE SUBGRAPH WEIGHTS

To optimize $\mathcal{L}^{\mathrm{CVAE_{corr}}}(\boldsymbol{\lambda}, \boldsymbol{\theta})$, we need to efficiently compute the weights $w_{G,e}^{\mathrm{MAS}}$ for each edge $e \in E$. These weights are only related to the graph $G$, but not the model distribution $p_{\boldsymbol{\theta}}$, the variational density functions $q_{\boldsymbol{\lambda}}$'s or the data $\boldsymbol{x}$.

Recall that $w_{G,e}^{\text{MAS}}$ is just the fraction of $G$'s maximal acyclic subgraphs which contain the edge $e$. For simplicity, we first consider a special case where $G$ is a connected graph. Then $w_{G,e}^{\text{MAS}}$ is just the fraction of $G$'s spanning trees which contain the edge $e$. To compute this quantity, we need to know the total number of spanning trees of $G$, as well as the number of spanning trees of $G$ which contain the edge $e$.

The **Matrix Tree Theorem** (Chaiken & Kleitman, 1978) gives a formula for the total number of spanning trees of a given graph $G$:

**Theorem 1** (Matrix Tree Theorem (Chaiken & Kleitman, 1978)). *For an undirected graph $G = (V, E)$, the number of spanning trees of $G$ is the determinant of the sub-matrix of the Laplacian matrix $L$ of $G$ after deleting the $i^{th}$ row and the $i^{th}$ column (i.e., the $(i,i)$-cofactor of $L$), for any $i = 1, ..., n$.*

The Laplacian matrix $L$ of a undirected graph $G = (V = \{v_1, \ldots, v_n\}, E)$ is defined as follows. $L \in \mathbb{R}^{n \times n}$ is a symmetric matrix where:

$$L_{i,j} = \begin{cases} \text{degree}(v_i) & \text{if } i = j, \\ -1 & \text{if } (v_i, v_j) \in E, \\ 0 & \text{otherwise.} \end{cases}$$

Similarly, we compute the number of spanning trees of $G$ that contain edge $e = (v_i, v_j)$:

**Theorem 2** (Number of spanning trees containing a specific edge). *For an undirected graph $G = (V, E)$ and an edge $(v_i, v_j) \in E$, the number of spanning trees of $G$ containing this edge is the determinant of the sub-matrix of the Laplacian matrix $L$ of $G$ after deleting the $i^{th}$, $j^{th}$ rows and the $i^{th}$, $j^{th}$ columns of it (i.e., the complement of the minor $M_{ij,ij}$ of $L$).*

*Proof.* See Appendix C.1. □

Directly using these two theorems to compute $w_{G,e}^{\text{MAS}}$ has several issues. First, we need to compute $|E|$ different determinants for matrices of size $(|V| - 2) \times (|V| - 2)$ (by applying Theorem 2), whose time complexity is $O(|E||V|^3)$, which is very inefficient. Second, the results we get from these two theorems can be exponentially large compared to $|V|$ and $|E|$ (e.g., a complete graph $K_n$ with $n$ vertices contains $n^{n-2}$ spanning trees), which can result in significant numerical issues under division. Since we only care about the ratios, notice that the numbers we compute from Theorem 2 are cofactors of the matrices (sub-matrices of $L$) on which we compute determinants from Theorem 1, we derive the following formula for computing the weights $w_{G,e}^{\text{MAS}}$'s, using the relationship between cofactors, determinants, and inverse matrices.

**Theorem 3** (Computing the spanning tree edge weights). *For an undirected connected graph $G = (V, E)$ and an edge $e = (v_i, v_j) \in E$, the weight $w_{G,e}^{MAS} = L_{i,i}^+ - L_{i,j}^+ - L_{j,i}^+ + L_{j,j}^+$. Here $L^+$ is the **Moore-Penrose inverse** of the Laplacian matrix $L$ of $G$.*

*Proof.* See Appendix C.2. □

With this theorem, given an undirected connected graph $G$, we can first compute the Moore-Penrose inverse $L^+$ of $G$'s Laplacian matrix and then compute the weight $w_{G,e}^{\text{MAS}}$ for every edge $e \in E$. The time complexity is $O(|V|^3)$, which is not unreasonable, even for relatively large graphs. Furthermore, there should be no numerical issues in the computations since computing the Moore-Penrose inverse is numerically stable.

We have just illustrated how to compute the weights $w_{G,e}^{\text{MAS}}$ when $G$ is connected. When $G$ is not connected, from the definition of maximal acyclic graphs, we know that, $w_{G,e}^{\text{MAS}}$ is equal to $w_{\text{CC}(G,e),e}^{\text{MAS}}$, where $\text{CC}(G,e)$ is the connected component of $G$ that contains the edge $e$. Therefore, we just need to apply Theorem 3 for all connected components of $G$. The details of computing the weights $w_{G,e}^{\text{MAS}}$ for all $e \in E$ are shown in Algorithm 1 in appendix. The time complexity of this algorithm is $O(|V|^3)$ in the worst case, but potentially much smaller if each connected component of $G$ has a small number of vertices. We can in fact relax the premise of Theorem 3 that $G$ is connected: since the Moore-Penrose inverse of block diagonal matrices is equivalent to computing the Moore-Penrose inverse for each

of these sub-matrices, Theorem 3 is also correct for general graphs. Hence, we can compute the weights without identifying the connected components. However, performing Algorithm 1 is at least as efficient as directly computing the Moore-Penrose inverse of the whole matrix.

### 3.4 REGULARIZATION WITH NON-EDGES

With Algorithm 1, we can efficiently compute all the weights $w_{G,e}^{\text{MAS}}$ and optimize the ELBO $\mathcal{L}^{\text{CVAE}_{\text{corr}}}$ in Eq. 9. This ELBO may be a good objective function to optimize if our goal is only to use the trained generative model $p_{\boldsymbol{\theta}}(\boldsymbol{x}|\boldsymbol{z})$ or to get a good approximation to the singleton and pairwise marginal posterior. However, if we want to use the learned pairwise variational density functions $q_{\boldsymbol{\lambda}}(\cdot,\cdot|\cdot,\cdot)$ for predictive tasks that may take inputs $(\boldsymbol{x}_u, \boldsymbol{x}_v)$ where $(u, v) \notin E$ (e.g., perform link predictions using these density functions), then purely optimizing Eq. 9 is not sufficient since this loss function may consider all pairs of vertices as correlated, due to only "positive examples" (correlated pairs) are present in the data. As a result, the learned pairwise density functions are not capable of identifying the correlations of new inputs.

To address this issue, we add a regularization term to the ELBO $\mathcal{L}^{\text{CVAE}_{\text{corr}}}$, which is akin to negative sampling in language modeling. Recall in Eq. 9, we compute the average KL terms over all maximal acyclic subgraphs of $G$. These KL terms are the "positive" examples. For the "negative" examples, we apply the average KL terms over all maximal acyclic subgraphs of the complete graph $K_n$ as the graph, and treat the prior on $\boldsymbol{z}$ to be i.i.d. on each $\boldsymbol{z}_i$ to regularize the density functions $q$'s towards independent for the negative samples. Since $K_n$ is a complete graph, the edge weight $w^{\text{MAS}}$ should be the same among all edges. By Proposition 1, since $K_n$ is connected and has $\frac{n(n-1)}{2}$ edges, we get

**Proposition 2** (Maximal acyclic subgraph weights for complete graphs). *For a complete graph $K_n$, the weight $w_{K_n,e}^{MAS}$ for any edge $e$ of $K_n$ is $\frac{2}{n}$.*

Therefore, we can define the loss function for $\text{CVAE}_{\text{corr}}$ with the negative sampling regularization as

$$\mathcal{L}^{\text{CVAE}_{\text{corr}}\text{-NS}}(\boldsymbol{\lambda}, \boldsymbol{\theta}) := \mathcal{L}^{\text{CVAE}_{\text{corr}}}(\boldsymbol{\lambda}, \boldsymbol{\theta}) - \gamma \cdot \Big( \sum_{i=1}^{n} \text{KL}(q_{\boldsymbol{\lambda}}(\boldsymbol{z}_i|\boldsymbol{x}_i)||p_0(\boldsymbol{z}_i))$$
$$+ \frac{2}{n} \sum_{1 \leq i < j \leq n} \mathbb{E}_{q_{\boldsymbol{\lambda}}(\boldsymbol{z}_i, \boldsymbol{z}_j|\boldsymbol{x}_i, \boldsymbol{x}_j)} \log \frac{q_{\boldsymbol{\lambda}}(\boldsymbol{z}_i, \boldsymbol{z}_j|\boldsymbol{x}_i, \boldsymbol{x}_j)}{q_{\boldsymbol{\lambda}}(\boldsymbol{z}_i|\boldsymbol{x}_i)q_{\boldsymbol{\lambda}}(\boldsymbol{z}_j|\boldsymbol{x}_j)} \Big). \quad (10)$$

Here $\gamma > 0$ is a parameter that controls the regularization. In this loss function, the edges in $E$ appear in both the positive and the negative samples. However, by Proposition 1, the average weight of the edges in $E$ in the positive samples is $\frac{|V|-|CC(G)|}{|E|}$. Therefore, as long as $\gamma \leq O\left(\frac{|V|(|V|-|CC(G)|)}{|E|}\right)$, the regularization term will not dominate the effect of the positive samples.

This negative sampling regularization can help $\text{CVAE}_{\text{corr}}$ learn better latent embeddings for many predictive tasks as shown in Section 4. $\text{CVAE}_{\text{ind}}$ does not need such regularization since it does not learn correlated variational density functions, but only fully-factorized ones.

## 4 EXPERIMENTS

We test the performance of $\text{CVAE}_{\text{ind}}$ and $\text{CVAE}_{\text{corr}}$ on different tasks on three datasets, and compare their performances to the baseline methods.

### 4.1 EXPERIMENT SETTINGS

**Tasks.** We test our methods on 3 tasks: user matching on a public movie rating dataset, spectral clustering on a synthetic tree-structured dataset and link prediction on a public product rating dataset. For each dataset, we have a high-dimensional feature for each user (or data point) and a undirected correlation graph between the users (or data points).

**Baselines.** We have two baseline methods:

- The standard Variational Auto-Encoders.

Table 1: Synthetic user matching test RR

| Method | Test RR |
|--------|---------|
| VAE | $0.3498 \pm 0.0167$ |
| CVAE$_{\text{ind}}$ | $0.6608 \pm 0.0066$ |
| CVAE$_{\text{corr}}$ | $\mathbf{0.7129 \pm 0.0096}$ |

- The GraphSAGE algorithm (Hamilton et al., 2017): the state-of-the-art method on learning latent embeddings with graph convolutional networks. It is capable of learning latent embeddings that take the correlation structures into consideration.

**Evaluations.** Different tasks may have different evaluation metrics. However, all of our results are computed based on the (expected) quadratic pairwise distance of the latent distributions on the evaluation dataset. For VAE, CVAE$_{\text{ind}}$ and CVAE$_{\text{corr}}$, denote the evaluation dataset as $\boldsymbol{x}_1, \ldots, \boldsymbol{x}_n$ and the marginal variational distribution on $(\boldsymbol{z}_i, \boldsymbol{z}_j)$ as $q(\boldsymbol{z}_i, \boldsymbol{z}_j)$, then the distance between the data points $i$ and $j$ ($i \neq j$) is defined as $\text{dis}_{i,j} = \mathbb{E}_{q(\boldsymbol{z}_i, \boldsymbol{z}_j)}\left[||\boldsymbol{z}_i - \boldsymbol{z}_j||_2^2\right]$. Notice that, for VAE and CVAE$_{\text{ind}}$, the distribution $q(\boldsymbol{z}_i, \boldsymbol{z}_j) = q(\boldsymbol{z}_i)q(\boldsymbol{z}_j)$ is always factorized. This does not hold for the CVAE$_{\text{corr}}$. For GraphSAGE, since the learned embeddings $\boldsymbol{z}_i$'s are not stochastic, we use $\text{dis}_{i,j} = ||\boldsymbol{z}_i - \boldsymbol{z}_j||_2^2$ as the distance between the data points $i$ and $j$.

Additional information related to the models and inference settings are in Appendix E.

## 4.2 RESULTS

### 4.2.1 USER MATCHING

We evaluate CVAE with a bipartite correlation graph. We use the **MovieLens 20M** dataset (Harper & Konstan, 2016). This is a public movie rating dataset that contains $\approx 138$K users and $\approx 27$K movies. We binarize the rating data and only consider whether a user has watched a movie or not, i.e., the feature vector for each user is a binary bag-of-word vector, and we only keep ratings for movies that have been rated at least 1000 times. For all the experiments, we did a stochastic train/test split over users with a $90/10$ ratio.

For each user $u_i$, we randomly split the movies that this user has watched into two halves and construct two synthetic users $u_i^A$ and $u_i^B$. This creates a bipartite graph where we know the synthetic users which were generated from the same real user should be more related than two random synthetic users. The goal of the evaluation is that, when given the watch history of a synthetic user $u_i^A$ from a held-out set, we try to identify its dual user $u_i^B$. This can be potentially helpful with identifying close neighbors when using matching to estimate causal effect, which is generally a difficult task especially in high-dimensional feature spaces (Imbens & Rubin, 2015).

We train all the methods on all the synthetic user pairs from the training set. To evaluate, we select a fixed number of $N^{\text{eval}} = 1000$ pairs of synthetic user from the test sets. For each synthetic user $u_i^A$ (or $u_i^B$), we find the ranking of $u_i^B$ (or $u_i^A$) among all candidates in the set of all other $2N^{\text{eval}} - 1$ synthetic users in terms of the latent embedding distance to $u_i^A$ (or $u_i^B$). The latent embedding distance metrics $\text{dis}_{i,j}$'s for all methods are defined in Section 4.1. For CVAE$_{\text{corr}}$, we set the negative sampling regularization parameter $\gamma = 1$.

We report the average Reciprocal Rank (RR) of the rankings for all methods in Table 1. CVAE$_{\text{ind}}$ and CVAE$_{\text{corr}}$ strongly outperform the standard VAE, which means that the correlation structure helps in learning useful latent embeddings. Here CVAE$_{\text{corr}}$ improves over CVAE$_{\text{ind}}$ by learning a correlated variational approximation. We do not compare our methods to the GraphSAGE algorithm for this task since the graph for this task is just a bipartite matching graph with many connected components while GraphSAGE works well for graphs where the local neighborhoods can provide substantial information for the vertices.

### 4.2.2 SPECTRAL CLUSTERING

We perform spectral clustering (Von Luxburg, 2007) on a synthetic dataset with a tree-structured latent variable graphical model. The dataset contains $N = 10000$ data points $x_1, \ldots, x_N \in \mathbb{R}^D$ with $D = 1000$. Each $x_i$ is generated independently from the distribution $p(x_i|z_i)$ given the ground truth latent embeddings $z_1, \ldots, z_N \in \mathbb{R}^d$, where the likelihood $p(\cdot|\cdot)$ is element-wise Bernoulli distribution with the logits come from a two-layer feedforward neural network that takes the latent embeddings as inputs.

The latent embeddings $z_1, \ldots, z_n$ are drawn from an tree-structured undirected graphical model. The probability distribution of this graphical model is of the same form as Eq. 4, where the singleton prior density $p_0(\cdot)$ is the standard normal and the pairwise prior density is $p_0(\cdot, \cdot) = \mathcal{N}\left(\mu = \mathbf{0}_{2d}, \Sigma = \begin{pmatrix} I_d & \tau \cdot I_d \\ \tau \cdot I_d & I_d \end{pmatrix}\right)$. With the latent embeddings $z_1, \ldots, z_N$, we generate a binary cluster label $c_i \in \{0, 1\}$ for each data point $x_i$ by performing a principle components analysis on the latent embeddings and set $c_i = 1$ if and only if $z_i$ has a coefficient rank at least $\frac{N}{2}$ among all the $N$ data points on the first component.

We perform spectral clustering based on the latent embedding distance metrics $\text{dis}_{i,j}$'s discussed in Section 4.1 for all algorithms. Since spectral clustering requires a non-negative symmetric similarity matrix $S \in \mathbb{R}^{N \times N}$, we set $S_{ij} = \exp\left(-\text{dis}_{i,j}/2\right)$. We apply a normal spectral clustering procedure by computing the eigenvector $v \in \mathbb{R}^N$ corresponding to the second smallest eigenvalue of the normalized Laplacian matrix of $S$. Then we cluster the data points $x_1, \ldots, x_n$ by clustering the set of coordinates of $v$ with value larger than the median of all these coordinates as one cluster, and the rest as the other cluster.

To evaluate clustering, we apply the normalized mutual information score (Vinh et al., 2010), which is in the range of $[0, 1]$ (larger the better). The scores for all methods are shown in Table 2. Here we apply negative sampling regularization parameter $\gamma = 0.1$ for $\text{CVAE}_{\text{corr}}$. From Table 2, we can see that $\text{CVAE}_{\text{ind}}$ and $\text{CVAE}_{\text{corr}}$ strongly outperform both baseline methods. $\text{CVAE}_{\text{corr}}$ does not significantly improve over $\text{CVAE}_{\text{ind}}$ potentially due to that we do not have sufficiently many edges to learn a good correlation function, but it still outperforms the baseline methods.

Table 2: Spectral clustering normalized mutual information scores

| Method | MI scores |
| --- | --- |
| VAE | $0.0031 \pm 0.0059$ |
| GraphSAGE | $0.0945 \pm 0.0607$ |
| $\text{CVAE}_{\text{ind}}$ | $\mathbf{0.2821 \pm 0.1599}$ |
| $\text{CVAE}_{\text{corr}}$ | $\mathbf{0.2748 \pm 0.0462}$ |

Table 3: Link prediction test Normalized CRR

| Method | Test NCRR |
| --- | --- |
| VAE | $0.0052 \pm 0.0007$ |
| GraphSAGE | $0.0115 \pm 0.0025$ |
| $\text{CVAE}_{\text{ind}}$ | $0.0160 \pm 0.0004$ |
| $\text{CVAE}_{\text{corr}}$ | $\mathbf{0.0171 \pm 0.0009}$ |

### 4.2.3 LINK PREDICTION

We perform link prediction on a general undirected graph $G = (V, E)$. In this experiment, we use the Epinions dataset (Massa & Avesani, 2007), which is a public product rating dataset that contains $\approx 49$K users and $\approx 140$K products. We again binarize the rating data and create a bag-of-words binary feature vector for each user. We only keep products that have been rated at least 100 times and only consider users who have rated these products at least once.

We construct a correlation graph $G = (V, E)$. The Epinions dataset has a set of single-directional "trust" statements between the users, i.e., a directed graph $G' = (V', E')$ among all users. Since we need undirected correlation structures, we take all vertices from $G'$ (i.e., set $V = V'$) and set an edge $(v_i, v_j) \in E$ if both $(v_i', v_j')$ and $(v_j', v_i')$ are in $E'$.

To split the train/test dataset for the link prediction task, for each vertex $v_i \in V$, we hold out $\max\left(1, \frac{1}{20} \cdot \text{degree}(v_i)\right)$ edges on $v_i$ as the testing edge set $E_{\text{test}}$, and put all edges that are not selected into the training edge set $E_{\text{train}}$. We train all methods on the product ratings and the correlation graph $G^{\text{train}} = (V, E_{\text{train}})$.

For evaluation, we first compute the latent embedding distance $\text{dis}_{i,j}$'s that was defined in Section 4.1 for $1 \leq i \neq j \leq N$. Then for each user $u_i$, we compute the Cumulative Reciprocal Rank $\text{NCRR}_i$ of the ratings of $u_i$'s testing edges, among all possible connections except for the edges in the training edge set, in terms of the latent embedding distance metrics. Formally, this value equals $\text{CRR}_i = \sum\limits_{(v_i, v_j) \in E_{\text{test}}} \frac{1}{|\{k:(v_i,v_k) \notin E_{\text{train}}, \text{dis}_{i,k} \leq \text{dis}_{i,j}\}|}$.

Larger $\text{CRR}_i$ indicates better ability to predict held-out links. We further normalize the CRR values to within the range of $[0, 1]$ and show the average metrics among all users in Table 3. Evidently, CVAE$_{\text{ind}}$ and CVAE$_{\text{corr}}$ again strongly outperform the baseline methods. Here we apply a large regularization value of $\gamma = 1000$ for the CVAE$_{\text{corr}}$, which may help when the input graph has a complex structure (unlike the previous experiments) yet does not have a dense connection. This choice of $\gamma$ provides CVAE$_{\text{corr}}$ enough regularization for learning the correlation and helps it improve over CVAE$_{\text{ind}}$.

## 5 RELATED WORK

Shaw et al. (2011) incorporated graph structure to metric learning. The major difference with CVAEs is that the metric learned from Shaw et al. (2011) is inherently linear while CVAEs are capable of capturing more complex non-linear relations in the feature space.

There has been some previous work on handling optimizations with intractability or inconsistency over general graphs by leveraging the problems on the spanning trees. In graphical model inference, the tree-reweighed sum-product algorithm (Wainwright, 2003) and the tree-reweighed Bethe variational principle (Wainwright et al., 2005) extend the ordinary Bethe variational principle (Wainwright & Jordan, 2005) over a convex combination of tree-structured entropies. In combinatorial optimization, Tang & Jebara (2017) used the maximal spanning trees of the pairwise matching scores to provide consistent initializations for multi-way matching.

Moreover, there has been some recent work on incorporating structures in variational inference and VAEs. Hoffman & Blei (2015) proposed structured stochastic variational inference to improve over the naive mean-field family. Johnson et al. (2016) proposed structured VAEs which enable the prior to take a more complex form (e.g., a Gaussian mixture model, or a hidden Markov model). Lin et al. (2018) extend the work of Johnson et al. (2016) to variational message passing, which is applied to analyze time-seres data in Pearce et al. (2018). Burda et al. (2016) and Kingma et al. (2016) proposed variational approximation families that can learn better posterior approximations compared to the commonly applied mean-field family, with normalizing flows and inverse autoregressive flows, respectively. Ainsworth et al. (2018) proposed output-interpretable VAEs which combine a structured VAE comprised of group-specific generators with a sparsity-inducing prior. The resent work StructVAE (Yin et al., 2018) extends the standard VAE to handle tree-structured latent variables. However, most of these works are designed to model the structures between dimensions *within* each data point, while CVAEs considers general graph structures *between* data points.

Another related line of work are the recent advances in convolutional networks for graphs (Bruna et al., 2015; Duvenaud et al., 2015; Defferrard et al., 2016; Niepert et al., 2016; Kipf & Welling, 2016) and the extensions, e.g., our baseline method GraphSAGE (Hamilton et al., 2017).

## 6 CONCLUSION

We introduce CVAE$_{\text{ind}}$ and CVAE$_{\text{corr}}$ to account for correlations between data points that are known *a priori*. They extend the standard VAEs by applying a correlated prior on the latent variables. Furthermore, CVAE$_{\text{corr}}$ adopts a correlated variational density function to achieve a better variational approximation. These methods successfully outperform the baseline methods on several machine learning tasks using the latent variable distance metrics. For future work, we will study the correlated variational auto-encoders that consider higher-order correlations.

### ACKNOWLEDGMENTS

This work was supported in part by National Science Foundation grant III: Small: Collaborative Research: Approximate Learning and Inference in Graphical Models III-1526914.

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

APPENDIX

In the appendix, we first show the ELBOs for the CVAEs under acyclic correlation graphs, as well as the derivation for the ELBO of CVAE$_\text{corr}$ (Eq. 9) for general correlation graphs. Also we show the counter example mentioned in Section 3.1. Then, we show the proofs of Theorems 2 and 3 and the details for Algorithm 1. At the end, we provide more information related to the models and inference settings.

## A  ELBOs AND DERIVATIONS

**ELBO for** CVAE$_\textbf{ind}$ **with acyclic graphs.**   CVAE$_\text{ind}$ applies a correlated prior $p_0^\text{corr}(z)$ as in Eq. 4. Therefore,

$$
\begin{aligned}
&\mathcal{L}^{\text{CVAE}_\text{ind}\text{-acyclic}}(\boldsymbol{\lambda}, \boldsymbol{\theta}) \\
=&\mathbb{E}_{q_{\boldsymbol{\lambda}}(\boldsymbol{z}|\boldsymbol{x})}\left[\log p_{\boldsymbol{\theta}}(\boldsymbol{x}|\boldsymbol{z})\right] - \text{KL}(q_{\boldsymbol{\lambda}}(\boldsymbol{z}|\boldsymbol{x})||p_0(\boldsymbol{z})) \\
=&\sum_{i=1}^{n}\left(\mathbb{E}_{q_{\boldsymbol{\lambda}}(\boldsymbol{z}_i|\boldsymbol{x}_i)}\left[\log p_{\boldsymbol{\theta}}(\boldsymbol{x}_i|\boldsymbol{z}_i)\right] - \text{KL}(q_{\boldsymbol{\lambda}}(\boldsymbol{z}_i|\boldsymbol{x}_i)||p_0(\boldsymbol{z}_i))\right) \\
&+ \sum_{(v_i,v_j)\in E}\mathbb{E}_{q_{\boldsymbol{\lambda}}(\boldsymbol{z}_i|\boldsymbol{x}_i)q_{\boldsymbol{\lambda}}(\boldsymbol{z}_j|\boldsymbol{x}_j)}\log\frac{p_0(\boldsymbol{z}_i,\boldsymbol{z}_j)}{p_0(\boldsymbol{z}_i)p_0(\boldsymbol{z}_j)}.
\end{aligned}
\tag{11}
$$

This ELBO is a lower bound of the log probability $p(\boldsymbol{x};\boldsymbol{\theta})$ under the correlated prior (as in Eq. 4). Optimizing this ELBO helps the variational distributions to regularize according to the correlation graph $G$.

**ELBO for** CVAE$_\textbf{corr}$ **with acyclic graphs.**   By changing $q$ to be the correlated distribution as in Eq. 5, the ELBO in Eq. 11 becomes

$$
\begin{aligned}
&\mathcal{L}^{\text{CVAE}_\text{corr}\text{-acyclic}}(\boldsymbol{\lambda}, \boldsymbol{\theta}) \\
=&\sum_{i=1}^{n}\left(\mathbb{E}_{q_{\boldsymbol{\lambda}}(\boldsymbol{z}_i|\boldsymbol{x}_i)}\left[\log p_{\boldsymbol{\theta}}(\boldsymbol{x}_i|\boldsymbol{z}_i)\right] - \text{KL}(q_{\boldsymbol{\lambda}}(\boldsymbol{z}_i|\boldsymbol{x}_i)||p_0(\boldsymbol{z}_i))\right) \\
&- \sum_{(v_i,v_j)\in E}\Big(\text{KL}(q_{\boldsymbol{\lambda}}(\boldsymbol{z}_i,\boldsymbol{z}_j|\boldsymbol{x}_i,\boldsymbol{x}_j)||p_0(\boldsymbol{z}_i,\boldsymbol{z}_j)) \\
&- \text{KL}(q_{\boldsymbol{\lambda}}(\boldsymbol{z}_i|\boldsymbol{x}_i)||p_0(\boldsymbol{z}_i)) - \text{KL}(q_{\boldsymbol{\lambda}}(\boldsymbol{z}_j|\boldsymbol{x}_j)||p_0(\boldsymbol{z}_j))\Big).
\end{aligned}
\tag{12}
$$

**Derivation for Eq. 9.**   We derive the ELBO of CVAE$_\text{corr}$ (Eq. 9) for general correlation graphs here. Recall that Eq. 7 shows that

$$
\log p_{\boldsymbol{\theta}}(\boldsymbol{x}) \geq \frac{1}{|\mathcal{A}_G|}\sum_{G'\in\mathcal{A}_G}\Big(\mathbb{E}_{q_{\boldsymbol{\lambda}}^{G'}(\boldsymbol{z}|\boldsymbol{x})}[\log p_{\boldsymbol{\theta}}(\boldsymbol{x}|\boldsymbol{z})] \\
- \text{KL}(q_{\boldsymbol{\lambda}}^{G'}(\boldsymbol{z}|\boldsymbol{x})||p_0^{G'}(\boldsymbol{z}))\Big).
$$

By the definition of $\mathcal{A}_G$, the right hand side of the above inequality equals the following sum

$$
\begin{aligned}
&\sum_{i=1}^{n}\left(\mathbb{E}_{q_{\boldsymbol{\lambda}}(\boldsymbol{z}_i|\boldsymbol{x}_i)}\left[\log p_{\boldsymbol{\theta}}(\boldsymbol{x}_i|\boldsymbol{z}_i)\right] - \text{KL}(q_{\boldsymbol{\lambda}}(\boldsymbol{z}_i|\boldsymbol{x}_i)||p_0(\boldsymbol{z}_i))\right) \\
&- \frac{1}{|\mathcal{A}_G|}\sum_{G'\in\mathcal{A}_G}\sum_{(v_i,v_j)\in E'}\Big(\text{KL}(q_{\boldsymbol{\lambda}}(\boldsymbol{z}_i,\boldsymbol{z}_j|\boldsymbol{x}_i,\boldsymbol{x}_j)||p_0(\boldsymbol{z}_i,\boldsymbol{z}_j)) \\
&- \text{KL}(q_{\boldsymbol{\lambda}}(\boldsymbol{z}_i|\boldsymbol{x}_i)||p_0(\boldsymbol{z}_i)) - \text{KL}(q_{\boldsymbol{\lambda}}(\boldsymbol{z}_j|\boldsymbol{x}_j)||p_0(\boldsymbol{z}_j))\Big)
\end{aligned}
$$

The pairwise sum part of the above equation is an average over a sum over all edges of all maximal acyclic subgraphs of $G$. Therefore, for each edge $e = (v_i, v_j) \in E$, the number of times it appears in this pairwise sum part of the above sum is just the number of maximal acyclic subgraphs containing

this edge. Therefore, this part can be viewed as a weighed sum over all edges in $E$, where the weights come from the fraction ratios in Definition 2. With this definition, we can further write the above sum as

$$\sum_{i=1}^{n} \Big( \mathbb{E}_{q_{\boldsymbol\lambda}}(\boldsymbol{z}_i|\boldsymbol{x}_i) [\log p_{\boldsymbol\theta}(\boldsymbol{x}_i|\boldsymbol{z}_i)] - \mathrm{KL}(q_{\boldsymbol\lambda}(\boldsymbol{z}_i|\boldsymbol{x}_i)||p_0(\boldsymbol{z}_i)) \Big)$$

$$- \sum_{(v_i,v_j)\in E} w^{\mathrm{MAS}}_{G,(v_i,v_j)} \Big( \mathrm{KL}(q_{\boldsymbol\lambda}(\boldsymbol{z}_i,\boldsymbol{z}_j|\boldsymbol{x}_i,\boldsymbol{x}_j)||p_0(\boldsymbol{z}_i,\boldsymbol{z}_j))$$

$$- \mathrm{KL}(q_{\boldsymbol\lambda}(\boldsymbol{z}_i|\boldsymbol{x}_i)||p_0(\boldsymbol{z}_i)) - \mathrm{KL}(q_{\boldsymbol\lambda}(\boldsymbol{z}_j|\boldsymbol{x}_j)||p_0(\boldsymbol{z}_j)) \Big)$$

$$= \mathcal{L}^{\mathrm{CVAE_{corr}}}(\boldsymbol{\lambda}, \boldsymbol{\theta}),$$

which is exactly Eq. 9.

## B    COUNTER EXAMPLE FOR SECTION 3.1

In Section 3.1, we mentioned that directly applying the ELBOs we derived (equations shown in Appendix A) will fail partly due to that optimizing over these ELBOs may lead the algorithm to learn useless parameters that lead the loss goes to infinity due to that these two equations are not guaranteed to be a lower bound of some log-likelihood function for general graphs. Here we provide a simple example to illustrate this.

Let us consider $G = (V, E)$ is a $K_4$ complete graph with $|V| = 4$ vertices and $\binom{|V|}{2} = 6$ edges. For simplicity, we consider the latent variables $z_1, z_2, z_3, z_4 \in \mathbb{R}$ and the prior distribution $p_0(z_i, z_j) = \mathcal{N}\left( \begin{pmatrix} z_i \\ z_j \end{pmatrix}; \boldsymbol{\mu} = \mathbf{0}_2, \Sigma = \begin{pmatrix} 1 & \tau \\ \tau & 1 \end{pmatrix} \right)$ for some $\tau \in (0, 1)$. If we extend the CVAE$_{\mathrm{ind}}$ and for the variational distribution, we set $q(z_i|\boldsymbol{x}_i)$ to be the normal distribution $N(\mu_i, \sigma_i^2)$, by simply apply Eq. 11, then the loss function becomes

$$\mathcal{L} = \sum_{i=1}^{4} \Big( \mathbb{E}_{q_{\boldsymbol\lambda}(z_i|\boldsymbol{x}_i)} [\log p_{\boldsymbol\theta}(\boldsymbol{x}_i|z_i)] + \mu_i^2 - \frac{1+2\tau^2}{2(1-\tau^2)}\sigma_i^2$$

$$+ \ln(\sigma_i) \Big) - \frac{1}{2(1-\tau^2)} \sum_{1\le i<j\le 4} (\mu_i^2 + \mu_j^2 - 2\tau\mu_i\mu_j).$$

If we maintain $\sigma_i$'s unchanged, set the model parameter $\boldsymbol{\theta}$ in a way that makes $p_{\boldsymbol\theta}(\boldsymbol{x}_i|z_i)$ unrelated to $z_i$ (e.g. set the parameter that multiply with $z_i$ to be 0) and set $\mu_1 = \mu_2 = \mu_3 = \mu_4 = \mu$ and let $\mu \to \infty$, then $\mathcal{L}$ will go to $+\infty$ if $\tau > \frac{1}{2}$. Therefore, directly applying Eq. 11 does not work. Directly applying Eq. 12 (i.e. extending the CVAE$_{\mathrm{corr}}$) will make the result even worse since it always has an optimal value at least as high as Eq. 11. In general, any general graphs with $K_4$ subgraph may suffer from the issue we just mentioned. We can not obtain useful latent embeddings by directly applying the Eqs. 11 and 12.

## C    PROOFS

We prove the Theorems 2 and 3 here.

### C.1    PROOF OF THEOREM 2

*Proof.* Given the graph $G = (V, E)$ and the edge $(v_i, v_j) \in E$. Denote $G$'s Laplacian matrix as $L$. Also denote $L_{-a,-b}$ as the sub-matrix of $L$ after deleting the $a^{th}$ row and the $b^{th}$ column. Denote $L_{-ab,-cd}$ as the sub-matrix of $L$ after deleting the $a^{th}$, $b^{th}$ rows and the $c^{th}$, the $d^{th}$ columns.

By Matrix Tree Theorem (Theorem 1 (Chaiken & Kleitman, 1978)), we know that the $(i, i)$-cofactor $C_{i,i} = |L_{-i,-i}|$ is the number of spanning trees of $G$.

Construct a graph $G' = (V, E/\{v_i, v_j\})$, i.e. the graph $G$ after removing the edge $(v_i, v_j)$. Denote the Laplacian matrix of $G'$ as $L'$. Then we will find that the matrix $L'_{-i,-i}$ is the same with $L_{-i,-i}$

except that they $L_{-i,-i}$ is 1 larger than $L'_{-i,-i}$ on the entry at $(j,j)$. By Matrix Tree Theorem, $|L'_{-i,-i}|$ is the number of spanning trees of $G'$. Since $G$ differs from $G'$ by only having one more edge $(v_i, v_j)$, we know that $|L_{-i,-i}| - |L'_{-i,-i}|$ represents the number of spanning trees in $G$ that contains the edge $(v_i, v_j)$.

Since we know that

$$
\begin{cases}
|L_{-i,-i}| & = \sum\limits_{k \neq i} (-1)^{j+k} |L_{-ij,-ik}| \\
|L'_{-i,-i}| & = \sum\limits_{k \neq i} (-1)^{j+k} |L'_{-ij,-ik}|.
\end{cases}
$$

. Subtract the second equation from the first one we get

$$
|L_{-i,-i}| - |L'_{-i,-i}| = (-1)^{2j} |L_{-ij,-ij}|
$$

which is just the complement of the Minor $M_{ij,ij}$ of $L$. Hence, the number of spanning trees of $G$ that contains the edge $(v_i, v_j)$ is $M_{ij,ij}$, the determinant of the sub-matrix of the Laplacian matrix $L$ of $G$, after deleting the the $i^{th}$, $j^{th}$ rows and the $i^{th}$, the $j^{th}$ columns. $\qquad\square$

### C.2 PROOF OF THEOREM 3

*Proof.* We borrow the notations from Appendix C.1. Given the undirected connected graph $G = (V, E)$ and an edge $(v_i, v_j) \in E$, we want to compute the ratio $w^{\text{MAS}}_{G,(v_i,v_j)}$. Since $G$ is connected, this ratio is just the fraction of $G$'s spanning trees containing the edge $(v_i, v_j)$. By Theorem 1 and 2, this ratio is just $\frac{|L_{-ij,-ij}|}{|L_{-i,-i}|}$.

Since $G$ is connected, it contains at least one spanning tree. Hence $|L_{-i,-i}| > 0$, which means $L_{-i,-i}$ is invertible. Denote the entry at $(j,j)$ of $L_{-i,-i}$ as $L_{-i,-i;j,j}$. Since the $L_{-i,-i}$ is invertible, we know that

$$
\frac{|L_{-ij,-ij}|}{|L_{-i,-i}|} = L^{-1}_{-i,-i;j,j}.
$$

Consider the original Laplacian matrix $L$ before deleting any row and column. Denote $|V| = n$. Since $L$ is always symmetric and always have an eigenvector $\boldsymbol{v}_n = \frac{1}{\sqrt{n}} \mathbf{1}_n$ with corresponding eigenvalue $\lambda_n = 0$, we perform eigenvalue decomposition on $L$ and write $L$ as:

$$
L = \sum_{k=1}^{n} \lambda_i \boldsymbol{v}_k \boldsymbol{v}_k^\top = \sum_{k=1}^{n-1} \lambda_i \boldsymbol{v}_k \boldsymbol{v}_k^\top.
$$

Where $\lambda_1, \ldots, \lambda_{n-1}, \lambda_n$ are $L$'s eigenvalues and $\boldsymbol{v}_1, \ldots, \boldsymbol{v}_{n-1}, \boldsymbol{v}_n$ are the corresponding orthogonal unit eigenvectors (i.e. $Q_v = (\boldsymbol{v}_1 \quad \cdots \quad \boldsymbol{v}_{n-1} \quad \boldsymbol{v}_n)$ is an orthogonal matrix).

Denote $v_{a,b}$ as the $b$-the coordinate of $\boldsymbol{v}_a$ and $\boldsymbol{v}_{a,-b}$ as the sub-vector $\boldsymbol{v}_a$ after deleting the $b^{th}$ coordinate. Then

$$
L_{-i,-i} = \sum_{k=1}^{n-1} \lambda_k \boldsymbol{v}_{k,-i} \boldsymbol{v}_{k,-i}^\top.
$$

$L_{-i,-i}$ is invertible, which is of rank $n-1$. While each of the matrix $\boldsymbol{v}_{k,-i} \boldsymbol{v}_{k,-i}^\top$ is of rank 1. Hence, we must have $\lambda_i \neq 0$ for all $i \in \{1, \ldots, n-1\}$.

Construct vectors $\boldsymbol{u}_1, \ldots, \boldsymbol{u}_n \in \mathbb{R}^n$ such that

$$
u_{k,a} = \begin{cases} v_{k,a} - v_{k,i} & \text{if } a \neq i \\ v_{k,a} & \text{if } a = i. \end{cases} \tag{13}
$$

Also, construct the matrix

$$
U = \sum_{k=1}^{n-1} \lambda_k^{-1} \boldsymbol{u}_k \boldsymbol{u}_k^\top.
$$

Since we know that $(\boldsymbol{v}_1 \quad \cdots \quad \boldsymbol{v}_{n-1} \quad \boldsymbol{v}_n)$ forms an orthogonal basis and $\boldsymbol{v}_n = \frac{1}{\sqrt{n}} \mathbf{1}_n$, it is easy to see (after simple calculations) that

$$
\boldsymbol{u}_{k,-i}^\top \cdot \boldsymbol{v}_{k',-i} = \begin{cases} 1 & \text{if } k = k' \\ 0 & \text{if } k \neq k'. \end{cases}
$$

Therefore, we will have

$$U_{-i,-i}L_{-i,-i} = I_{n-1}$$

which indicates that $U_{-i,-i} = L_{-i,-i}^{-1}$. Hence, the ratio we want to find is just $U_{-i,-i;j,j}$.

Recall the definition of $\boldsymbol{u}_1, \ldots, \boldsymbol{u}_n$ in Eq. 13, denote $Q_u = (\boldsymbol{u}_1 \quad \cdots \quad \boldsymbol{u}_{n-1} \quad \boldsymbol{u}_n)$, we get $Q_u = P_i Q_v$, where

$$P_i = \begin{pmatrix} 1 & & & & -1 & & & & \\ & 1 & & & -1 & & & & \\ & & \ddots & & \vdots & & & & \\ & & & 1 & & & & & \\ & & & & -1 & 1 & & & \\ & & & & \vdots & & \ddots & & \\ & & & & -1 & & & 1 & \\ & & & & -1 & & & & 1 \end{pmatrix} \in \mathbb{R}^{n \times n}$$

where the $-1$'s appear on the $i^{th}$ column. Hence, denote $D = \mathrm{diag}(\lambda_1, \ldots, \lambda_n)$, we have

$$U = Q_u D^+ Q_u^\top = P_i Q_v D^+ Q_v^\top P_i^\top = P_i L^+ P_i^\top.$$

Therefore, the ratio $w_{G,(v_i,v_j)}^{\mathrm{MAS}}$ we want to find, which is equal to $U_{-i,-i;j,j}$, is just the $(j,j)$-entry of $P_i L^+ P_i^\top$, which is

$$L_{i,i}^+ - L_{i,j}^+ - L_{j,i}^+ + L_{j,j}^+.$$

$\square$

## D ALGORITHM FOR COMPUTING THE WEIGHTS $w_{G,e}^{\mathrm{MAS}}$

Given a correlation graph $G = (V, E)$, we show the details of computing the weights $w_{G,e}^{\mathrm{MAS}}$ for all $e \in E$ as in Algorithm 1.

---

**Algorithm 1** Computing all weights $w_{G,e}^{\mathrm{MAS}}$

---

**Input:** undirected graph $G = (V = \{v_1, \ldots, v_n\}, E)$.
Compute all the connected components $\mathrm{CC}_1, \ldots, \mathrm{CC}_K$ of $G$ using depth-first search or breadth-first search.
**for** $k = 1$ **to** $K$ **do**
    Compute the Moore-Penrose inverse $L_k^+$ of the Laplacian matrix $L_k$ of the component $\mathrm{CC}_k$.
    Apply Theorem 3 to compute $w_{G,e}^{\mathrm{MAS}}$ for each edge $e$ in the component $\mathrm{CC}_k$.
**end for**
**Return** The weights $w_{G,e}^{\mathrm{MAS}}$ for all $e \in E$.

---

## E ADDITIONAL EXPERIMENT SETTINGS

We provide some additional information related to the models and inference settings here.

For all methods, we set the latent embeddings to have the dimensionality $d = 100$.

For the standard VAEs, the CVAE$_{\mathrm{ind}}$ and the CVAE$_{\mathrm{corr}}$, we apply a two-layer feed-forward neural network for the generative model $p_{\boldsymbol{\theta}}(\boldsymbol{x}_i|\boldsymbol{z}_i)$ and a two-layer feed-forward neural network for the singleton variational approximations $q_{\boldsymbol{\lambda}}(\boldsymbol{z}_i|\boldsymbol{x}_i)$. The model likelihood functions $p_{\boldsymbol{\theta}}(\boldsymbol{x}|\boldsymbol{z})$'s are Multinomial distribution (for the user matching and link prediction experiments) and Bernoulli distribution (for the spectral clustering experiments). The singleton variational approximations $q_{\boldsymbol{\lambda}}(\boldsymbol{z}_i|\boldsymbol{x}_i)$ are all diagonal normal distributions. The singleton prior density function $p_0(\cdot)$ is the standard normal distribution.

For the CVAEs, we set the pairwise prior density function $p_0(\cdot) = \mathcal{N}\left(\boldsymbol{\mu} = \boldsymbol{0}_{2d}, \Sigma = \begin{pmatrix} I_d & \tau \cdot I_d \\ \tau \cdot I_d & I_d \end{pmatrix}\right)$ for $\tau = 0.99$. It can be seen that, the singleton prior

density function $p_0(\cdot)$ and the pairwise prior density function $p_0(\cdot, \cdot)$ satisfy the constraints in Eq. 2. For the CVAE$_{\text{corr}}$, we treat $q_{\boldsymbol{\lambda}}(\boldsymbol{z}_i, \boldsymbol{z}_j | \boldsymbol{x}_i, \boldsymbol{x}_j)$ as a multivariate normal distributions such that the covariance matrices $\text{Cov}(\boldsymbol{z}_i, \boldsymbol{z}_i)$, $\text{Cov}(\boldsymbol{z}_j, \boldsymbol{z}_j)$ and $\text{Cov}(\boldsymbol{z}_i, \boldsymbol{z}_j)$ are all diagonal matrices. Instead of only learning the singleton density function $q_{\boldsymbol{\lambda}}(\cdot | \cdot)$, the CVAE$_{\text{corr}}$ also learn a two-layer feedforward neural network that takes the concatenation $(\boldsymbol{x}_i, \boldsymbol{x}_j)$ as input and output the covariance between $\boldsymbol{z}_i$ and $\boldsymbol{z}_j$ on each of these $d$ dimensions. As a result, $q_{\boldsymbol{\lambda}}(\boldsymbol{z}_i, \boldsymbol{z}_j | \boldsymbol{x}_i, \boldsymbol{x}_j)$ can be factorized as a product of $d$ bi-variate normal distributions, whose marginal distributions on $\boldsymbol{z}_i$ and $\boldsymbol{z}_j$ are consistent with the singleton variational approximations $q_{\boldsymbol{\lambda}}(\boldsymbol{z}_i | \boldsymbol{x}_i)$ and $q_{\boldsymbol{\lambda}}(\boldsymbol{z}_j | \boldsymbol{x}_j)$, respectively.

For GraphSAGE, we choose to use $K = 2$ aggregation steps and use the mean aggregator function. We use $Q = 20$ negative samples to optimize the loss function.

For all methods, we apply stochastic gradient optimizations with a step size of $10^{-3}$. We use the Adam algorithm (Kingma & Ba, 2015) to adjust the learning rates. All methods involve with stochastic batches with singleton terms. For these terms, we use a batch size $B_1 = 64$. For the CVAEs, there are some pairwise terms of the sum involve sampling edges from the graph $G = (V, E)$ (i.e. the "positive sampling") or sampling edges from the complete graph $K_n$ (i.e. the "negative sampling"). We use a batch size $B_2 = 256$ for sampling these pairwise terms.

