# OpenReview forum: "Correlated Variational Auto-Encoders"
_ICLR.cc/2019/Workshop/DeepGenStruct — DeepGenStruct 2019_

### Official Review · AnonReviewer2 · 2019-04-14
**Decent paper on extending the VAE framework with derived correlated prior from graph structures**

**Rating:** 4
**Confidence:** 2

**Review:**

The paper is based on the idea of extending the VAE framework beyond the simplistic IID assumption that it makes on the data. They do so by using structured priors that explicitly take the correlations between data points into account. The methods can be used to learn meaningful representation directly from a weighted graph of the data as shown in the experiments section. The only downside is that the paper does not compare or provide any detail of how their method relates to hierarchical Bayesian models such as LDA that also capture global level correlational statistics in categorical data.

Minor:
Sentences repeated twice at the end of page 2.

---

### Official Review · AnonReviewer1 · 2019-04-14
**Interesting and thorough work.**

**Rating:** 5
**Confidence:** 2

**Review:**

In this paper, the authors investigate generalizing VAEs to problems where correlations can be found between data points. They leverage known results (namely, writing the  distribution on tree-shaped graphical models as a function of pairwise joint distributions and marginals; for general graphs, they use mixture of tree distributions in a fashion reminiscent of tree-reweighted belief propagation) from graphical models to derive tractable evidence lower bound for the problem of interest.
They report improved results on a variety of datasets (spectral clustering, collaborative filtering and link prediction) compared to vanilla VAEs and a graph net based approach.

This is a good paper; the ideas are original, technically interesting, and well presented (there are some issues with language but nothing distracting), and results are convincing.

There was a slight missed opportunity in explaining in further details how the new bound interacted with sampling from the posterior (this is currently hidden in appendix D: the authors leverage properties of gaussian distributions; in general it would be more complicated to sample from the joint knowing only pairwise/marginals).

---

### Decision · Program_Chairs · 2019-04-19
**Acceptance Decision**

Accept